# Milnacipran Has an Antihyperalgesic Effect on Cisplatin-Induced Neuropathy

**DOI:** 10.3390/pharmaceutics15092218

**Published:** 2023-08-27

**Authors:** Sun Jin Cho, Jin Young Lee, Yujin Jeong, So Yeon Cho, Do-Gyeong Lee, Ji Yeon Choi, Hue Jung Park

**Affiliations:** 1Department of Anesthesiology and Pain Medicine, College of Medicine, The Catholic University of Korea, Seoul 06591, Republic of Korea; painezer1004@naver.com (S.J.C.); yujinjmd@gmail.com (Y.J.); ju-nu7@naver.com (S.Y.C.); dlehrud02@gmail.com (D.-G.L.); choi940207@gmail.com (J.Y.C.); 2Department of Anesthesiology and Pain Medicine, Samsung Medical Center, College of Medicine, The Sungkyunkwan University of Korea, Seoul 06351, Republic of Korea; l7035@hanmail.net

**Keywords:** allodynia, cisplatin, milnacipran, neuropathy, pain

## Abstract

(1) Background: Milnacipran is a typical serotonin–norepinephrine reuptake inhibitor and has been shown to have analgesic effects in several pain models. However, its antihyperalgesic effect in cisplatin-induced neuropathy remains unknown. We examined the effects of intraperitoneal (IP) milnacipran on allodynia in cisplatin-induced peripheral neuropathic mice. (2) Methods: Peripheral neuropathy was induced by injecting cisplatin (2.3 mg/kg/day, IP) six times, on every other day. Saline or milnacipran (10, 30, 50 mg/kg, IP) were then administered to the neuropathic mice. We examined mechanical allodynia using von Frey hairs at preadministration and at 30, 60, 90, 120, 180, 240 min and 24 h after drug administration. We also measured the dorsal root ganglion (DRG) activating transcription factor 3 (ATF3) to confirm the analgesic effects of milnacipran. (3) Results: For the milnacipran groups, the decreased paw withdrawal thresholds to mechanical stimuli were significantly reversed when compared to the preadministration values and the values in the saline-injected control group (*p* < 0.0001). Milnacipran administration to cisplatin-induced peripheral neuropathic mice resulted in a significant suppression of neuronal ATF3 activation (*p* < 0.01). (4) Conclusions: Milnacipran given via IP injection attenuates mechanical allodynia in mouse models of cisplatin-induced poly-neuropathic pain. These effects were confirmed by significant suppression of neuronal ATF3 activation in the DRG.

## 1. Introduction

Platinating agents, including cisplatin, are critical components of first-line chemotherapy for several cancers, such as ovarian, testicular, bladder, endometrial, head and neck, pancreatic, and breast cancers. Specifically, for germ cell tumors, cisplatin is instrumental in achieving a 5-year survival rate of over 95%. However, cisplatin also has certain adverse effects, such as neurotoxicity, ototoxicity, cardiometabolic side effects, and nephrotoxicity. Regrettably, there are no Food and Drug Administration (FDA)-approved drugs or preventive medications that address these toxicities [1,2,3,4].

Dolan et al. previously reported that symptoms of peripheral sensory neuropathy appear in 56% of testicular cancer patients treated with cisplatin chemotherapy [5]. Chemotherapy-induced peripheral neuropathy (CIPN) is a debilitating and painful condition that follows cancer chemotherapy. Symptoms include autonomic abnormalities (uncontrolled blood pressure, intestinal motility, and thermoregulation), motor abnormalities (incoordination and weakness), and sensory abnormalities (allodynia, hyperalgesia, spontaneous pain, and paresthesia) [6,7]. The worsening of CIPN can prompt a reduction in dosage or discontinuation of chemotherapy [8,9,10]. Therefore, management of CIPN is an important challenge in chemotherapy. However, the exact mechanism of CIPN was still unknown [11].

Milnacipran is a typical serotonin (5-hydroxytryptamine, 5-HT)–norepinephrine (NE) reuptake inhibitor (SNRI) and has shown analgesic effects in several pain models. By inhibiting 5-HT and NE reuptake, milnacipran has been demonstrated to alleviate chronic pain conditions such as fibromyalgia by modulating neurotransmitter levels in descending central nervous system inhibitory pathways [12]. Several studies have reported that the administration of milnacipran significantly reduces mechanical allodynia in mouse models, such as the paclitaxel-induced neuropathic pain model, formalin test model, chronic constriction injury model, spinal nerve ligation model, and streptozotocin-induced diabetic neuropathy model [13,14,15,16]. Although duloxetine, as an SNRI, has been reported to cause reduction in CIPN, the antihyperalgesic effect of milnacipran, one of the SNRIs, on cisplatin-induced neuropathy remains unexplored [17].

Neuropathic pain is associated with chronic inflammation and oxidative stress [18,19]. Activating transcription factor 3 (ATF3) is also an indicator of inflammation [20,21]. Therefore, in the present study, we quantified ATF3 in the dorsal root ganglion (DRG) to prove the presence or absence of the effect of the milnacipran in cisplatin-induced allodynia. Previous studies evaluated spinal cord or DRG for sensory nerve injury in cisplatin-induced mechanical allodynia mice. ATF3 in DRG was significantly increased in cisplatin-induced mechanical allodynia mice, but spinal glial astrocytes and microglia did not. Therefore, in the present study, we quantified ATF3 in DRG for the assessment of cisplatin-induced neuropathy [22].

In this study, we examined the effects of intraperitoneal (IP) milnacipran on allodynia in a mouse model of cisplatin-induced peripheral neuropathy.

## 2. Materials and Methods

### 2.1. Animals

This experiment was approved by the Institutional Animal Care and Use Committee at the Catholic University of Seoul, Korea. A semi-pathogen free barrier zone was provided for the experiment at the Catholic Laboratory Animal Research Center. Wild type male C57BL/6 mice (weight, 25–30 g; age 10 weeks old) were used in this study. The mice were individually housed in standard plastic cages with soft bedding and exposed to an alternating 12 h light–dark cycle (onset time: 7:00 a.m.) in a temperature- and humidity-controlled room (22 °C ± 0.5 °C). All the behavioral tests and medication deliveries were performed during the light cycle. Food and water were always provided without limitation. Under the ethical principles, behavioral testing was carried out. After completion of the test, the mice were euthanized.

### 2.2. Cisplatin Injection

Cisplatin (2.3 mg/kg/day, Spectrum Chemical MFG., Gardena, CA, USA) was administered intraperitoneally in the mice six times, on every other day, for total dose of 13.8 mg/kg. Mice in the vehicle group received saline instead of cisplatin six times, on every other day. The protection of kidney and liver function was planned via maintaining hydration and using lactated Ringer’s solution (0.25 mL) subcutaneously between cisplatin injection days. This protocol was based on a previously validated treatment/dosing paradigm [22]. A foot withdrawal response to von Frey filaments was measured on day 15. If a response occurred under applied force of 0.6 g or less to the hind paw, the mouse was to have considered developed mechanical allodynia, and was used in subsequent experiments. Impairment in motor function or mobility was not shown in cisplatin-administered mice during the observation. There was no change in stepping reflexes, pinnae, placing, or blink reflexes.

### 2.3. Milnacipran Administration

Mice demonstrating cisplatin-induced neuropathy were randomly assigned to one of four groups before drug administration on day 15 from first cisplatin injection. Saline or milnacipran was administered to the neuropathic mice at a single time on the same day. The control group received normal saline 1 mL/kg (*n* = 5). In the three experimental groups, Milna10, Milna30, and Milna50 (*n* = 6 per group), milnacipran was administered via IP injection at doses of 10, 30, or 50 mg/kg, respectively. Over the process of this animal experiment, milnacipran-treated mice did not represent any impairment in mobility or motor function.

### 2.4. Behavioral Test

Mechanical allodynia was assessed by measuring the paw withdrawal threshold (PWT) in response to mechanical stimulation. The PWT was tested using a series of von Frey filaments (Semmes Weinstein von Frey aesthesiometer, Stoelting Co., Wood Dale, IL, USA) in the up and down method [23]. Mice were positioned in plastic containers (8–8–8 cm) with wire mesh-bottoms for assessment of threshold in grams to produce PWT. The mice were given half an hour to accommodate themselves to the experiment environment. Then, each filament was perpendicularly pressed onto the midplantar surface of the hind paw through a wire floor for 6–8 s in a slightly bent shape. Flinching or brisk withdrawal of the paw in response to the filament was considered a positive response [20,24]. Consecutive stronger stimulus was applied when a response after filament stimulus was absent, while flinch or withdrawal response led to the next weaker filament. The PWT tests were repeated 5 additional times as this up–down pattern. After assessment of thresholds in both hind paws, the mean of the two hind paws was reported as result. All tests were performed at fixed times (1:00 pm to 6:00 pm) by an experimenter blinded to drug administration and dosing to avoid confounds by circadian rhythms. During cisplatin IP treatment, the behavioral test for mechanical allodynia was performed just before the daily administration. For this milnacipran study, we assessed mechanical allodynia in mice prior to drug administration, and then again at 30, 60, 90, 120, 180, 240 min and 24 h post-administration. Positive responses were recorded within the von Frey filaments range of 2.44 to 4.31 (0.03–2.00 g). All measured values were recorded as the mean value of interpretations from each of the hind paws. Measured values were tabulated, and a previously published formula was used to calculate the 50% response threshold [23,25]. Rotarod testing (Acceler rota-rod for rats 7750; Ugo Basile, Comerio-Varese, Italy) was used to test the motor levels in the cisplatin-induced mice. The neuropathic mice were adapted to the revolving drums and adapted to manipulate to reduce any stress during the experiment. Three training sessions were performed on the revolving drums (10–15 rpm) for 2 days before the actual test. For our study, the mice which could withstand on the revolving drum minimally for 150 s were selected. A control performance time was calculated as the mean of 3 training times. The rotarod time was recorded at 30, 60, 90, 120, 180, 240 min, and 24 h after IP injection. Each rotarod test was carried out 3 times at 5 min gaps, and the means were recorded [26].

### 2.5. Immunohistochemistry

The levels of activating transcription factor 3 (ATF3) in the dorsal root ganglion (DRG) of the control group and Milna50 group were evaluated. We collected DRG sections from each group. Six mice were selected for the Milna50 group and six mice for the control group, for a total of 12 mice. The mice were anesthetized with Euthasol^©^ and intracardiac perfusion was performed with 0.9% normal saline and 4% paraformaldehyde in consecutive order. The L5 DRG was dissected for immunohistochemistry. The 4% paraformaldehyde was used for fixation of DRG, and 30% sucrose was taken for cryoprotection. In common immunohisto-chemical method, fixed tissue was sectioned and processed. Tissues were cut off as 10 μm thickness. DRG tissues were mounted on glass slides. Non-specific binding was restricted through incubation in 2% normal goat serum in phosphate-buffered saline with 0.3% Tri-ton X-100 followed by incubation with primary ATF3 antibody (generated in rabbit, 1:1000; Santa Cruz Biotechnology, Santa Cruz, CA, USA) overnight at 4 °C. Binding sites were visualized with anti-rabbit IgG antibodies conjugated with Alex-488 (1:500; Invitrogen, Carlsbad, CA, USA). Nuclei were stained with ToPro3 (1:500; Invitrogen). In each mouse, two slides were made from both sides of L5 DRG. (The minimum interval between two slides was 180 μm.) All images were captured using a Leica TCS SP5 confocal imaging system and quantified using Image-Pro Plus v 6.0 (Media Cybernetics, Inc., Rockville, MD, USA). The white scale bar on the image is 100 μm. ATF3 quantification measured the total integrated intensity of pixels in a standardized area divided by the total number of pixels in 6 mice. ATF3 data are expressed as percentage change of the corresponding control. The researcher was blinded about the experimental condition during quantification. Raw data were used for statistical analysis.

### 2.6. Statistical Analysis

Statistical analysis was performed using GraphPad Prism (version 5.0, GraphPad Software, San Diego, CA, USA). Data are expressed as the mean ± standard error. Time response data are expressed in PWT for mechanical stimulation. A 2-way analysis of variance (ANOVA) was used for comparison of the PWT between time courses and milnacipran doses. A *t*-test was used to compare ATF3 levels in analysis of group differences. A *p* value of <0.05 was considered to indicate statistical significance.

## 3. Results

### 3.1. Cisplatin Injection Produces Mechanical Allodynia

No mice died or showed complications due to the cisplatin treatment schedule. During a few days after starting cisplatin treatment, mice lost their body weight slightly, but this did not show any significance compared with saline-treated mice. But, they caught up with normal weight gain during the post-treatment phase. Mice in the vehicle group, which received normal saline, did not demonstrate significant changes in PWT over time. However, the PWT of mice treated with cisplatin decreased significantly after six injections over 12 days (*p* < 0.0001). Mice with peripheral neuropathy induced by cisplatin displayed prominent allodynia, while those in the vehicle group did not (Figure 1). This study showed that mice with peripheral neuropathy, which were given IP injection of cisplatin according to a schedule, demonstrated prominent mechanical allodynia, whereas the vehicle group treated with saline showed no difference in PWT. The cisplatin group already showed a significant PWT reduction compared to the vehicle group when tested on day 3 after just one cisplatin IP administration (*p* < 0.05). Over time, the dose of cisplatin administered to the mice increased, thus decreasing the PWT to lower levels. Although IP administration of cisplatin ended on day 12, it did not recover to pre administration PWT levels over time, and remained significantly lower until day 30 (*p* < 0.0001).

### 3.2. Cisplatin-Induced Mechanical Allodynia Is Reversed by Milnacipran

The mice in the control group, which received normal saline, showed no change in mechanical allodynia at any time point. The anti-allodynic effect of milnacipran was most potent 30 min after injection, and the increase in PWT was proportional to the dose of milnacipran. For the milnacipran groups, PWT to mechanical stimuli significantly increased compared to both the pre administration values and the values in the control group (*p* < 0.0001) (Figure 2). In Milna10 group, PWT was significantly increased at 30 min (*p* < 0.01) and 60 min (*p* < 0.001) after administration. But 90 min after administration, PWT in Mina10 group did not show significant difference. In Milna30 group, PWT was significantly increased at 30 min (*p* < 0.0001), 60 min (*p* < 0.0001), and 90 min (*p* < 0.0001) after administration. In Milna50 group, PWT was significantly increased at 120 min (*p* < 0.0001), 180 min (*p* < 0.01), and 240 min (*p* < 0.001) after administration. The effect of milnacipran was rapid, as seen by the values measured 30 min after administration. These features of milnacipran represented not only an antiallodynic effect and but also a high degree of absorption and utilization. The time when the effect of the administered milnacipran became insignificant was 90 min in the Milna10 group and 120 min in the Milna30 group. The effect of the administered milnacipran in the Milna50 group kept for 240 min, and the effect disappeared after 24 h.

### 3.3. Milnacipran Treatment in Cisplatin-Induced Neuropathy Decreased ATF3

In mice treated with the vehicle, few ATF3-positive cells were observed in the DRG. Cisplatin-treated mice exhibited higher activation of neuronal ATF3 than vehicle-treated mice (*p* < 0.01). The ATF3 levels in the DRG were significantly lower in the Milna50 group than in the saline-injected control group. Immunohistochemistry demonstrated that milnacipran significantly suppressed the activation of neuronal ATF3 in cisplatin-induced neuropathic mice (*p* < 0.01) (Figure 3).

## 4. Discussion

### 4.1. Outcomes

Peripheral neuropathy resulting from cisplatin treatment leads to persistent tactile allodynia in mice. Gabapentin, etanercept, and ketorolac have been evaluated for cisplatin-induced allodynia in a mouse model in a previous study. We experienced that gabapentin and morphine reversed the allodynia in cisplatin-induced peripheral neuropathy, although ketorolac or etanercept could not reversed the allodynia. Additionally, etanercept pretreatment had a preventive effect on early phase of mechanical allodynia [22]. Several studies have also reported on the effects of milnacipran in mice with mechanical allodynia of various causes [13,14,15,16]. However, the antihyperalgesic effect of milnacipran on cisplatin-induced neuropathy had not yet been explored before this study. We confirmed that a mouse model injected intraperitoneally with cisplatin produced decreased PWT, signifying the development of mechanical allodynia. And, the decreased PWT was reversed subsequently by milnacipran administration, proportionally to capacity. The decrease in allodynia was confirmed by a decrease in ATF3 in DRG with immunohistochemistry. ATF3 expression in the DRG increases with chronic inflammation and nerve injury models in vivo [27,28]. ATF3 is also considered a DRG marker for damage to sensory afferents [24]. Here, we tested ATF3 to confirm the antihyperalgesic effect of milnacipran on cisplatin-induced neuropathy. In cisplatin-treated mice, the control group treated with saline intraperitoneally showed elevated activation of ATF3, whereas the Milna50 group treated with milnacipran showed significantly reduced activation of ATF3. Therefore, these findings, obtained by using immunohistochemistry methods, clarified the effect of milnacipran on cisplatin-induced mechanical allodynia in mice.

### 4.2. Difference from Other Chemotherapy Agents

Cisplatin and other platinating agents, which are included as first-line therapies in many cancer treatments, are the most widely used cytotoxic drugs [29,30]. Neurotoxic chemotherapy drugs can induce neuropathic symptoms, typically initiated with dysesthesia and paresthesia [31,32]. A normal touch could also be perceived as painful, a condition referred to as allodynia [33]. A cisplatin dose of 8.1–12.2 mg/kg administered over intervals of 3 to 4 weeks can induce tactile allodynia and dysesthesia in humans. Our previous research revealed that total cisplatin doses of 13.8 mg/kg induced a state of allodynia in mice [22]. Katsuyama et al. reported the effect of milnacipran in other chemotherapies induced peripheral neuropathic mouse models. Paclitaxel-induced peripheral neuropathy was induced after administration once per day for 5 days (2 mg/kg, IP). Paclitaxel-induced mechanical allodynia was reduced under repeated administration of milnacipran (both 10 and 20 mg/kg, once per day, IP) for 5 days, although a single administration of milnacipran (20 mg/kg, IP) had no effect on allodynia. Vincristine-induced peripheral neuropathy was induced after administration once per day for 7 days (0.1 mg/kg, IP). Vincristine-induced mechanical allodynia also did not show effective change after single injection of milnacipran (40 mg/kg, IP). Vincristine-induced mechanical allodynia was reduced significantly after repeated administration of milnacipran for 7 days (20 or 40 mg/kg, once per day, IP) [13,34]. By contrast, in current cisplatin experiment, allodynia was alleviated immediately at 30 min after single administration of milnacipran, not after repeated IP administration. There are certainly some differences between cisplatin (platinum) and other chemotherapeutic agents. For example, in cisplatin-treated patients, upper extremity symptoms are more prominent, while symptoms in paclitaxel-induced neuropathy are prominent in lower extremity. These features may be related to the different mechanism of CIPN. The pathophysiology of cisplatin-induced neuropathy is possibly due to a direct effect on DRG neurons [33]. Paclitaxel-induced neuropathy may be correlated with microtubule injury and impaired transportation of nutrients to the nerves. Damage on the mitochondria of nerves is considered a potential mechanism of vincristine-induced neuropathy. Also, this discrepancy might be due to differences of the strains that were used in the experiments. Katsuyama et al. used male ddY-strain mice [13,34], and we used wild-type male C57BL/6 mice for this study.

### 4.3. Mechanism by Milnacipran in Cisplatin-Induced Neuropathy

Milnacipran, a selective SNRI, is one of three drugs approved by the FDA for fibromyalgia. In an in vivo model, cisplatin-induced neurotoxicity was reported to relate serotonergic signaling, likely involving the serotonin-receptor subtype 7 [35]. And, the effect of NE in neuropathic pain was reported that have direct inhibition through α2-adrenergic receptor. Acting on the locus coeruleus, NE also improves the function of an impaired descending noradrenergic inhibitory system [36]. The pharmacokinetics of milnacipran are characterized by a short half-life (8 h), high degree of absorption and utilization, and high plasma binding [37,38]. The peak plasma concentration is reached 0.5 to 4 h after oral intake [39]. Preclinical data show that 10 mg/kg and 40 mg/kg doses of milnacipran increase serotonin (5-HT) and norepinephrine (NE) levels in the brain by three- to four-fold [40]. Side effects of this drug include headaches, nausea, vomiting, dizziness, constipation, hot flushes, insomnia, hyperhidrosis, palpitations, hypertension, and dry mouth. In this experiment, mechanical allodynia in a cisplatin-induced neuropathic mouse model showed a greater attenuation as the dose of milnacipran increased. Because milnacipran has many side effects, further study of the proper dose may be needed in the future.

Fibromyalgia is a disease with a wide spectrum of chronic pain. The clinically recommended dose of milnacipran for fibromyalgia is 100 mg/day (50 mg twice daily) and can be increased up to 200 mg/day (100 mg twice daily), depending on the patient’s specific needs. The inhibition of 5-HT and NE reuptake can decrease pain transmission [12]. Persistent pain induces changes in sensitivity within both ascending and descending nerve pathways in the brain and spinal cord [41]. Prolonged and sustained nerve injury and tissue damage stimulate nociceptive afferents, leading to central neuronal hyperexcitability. Neuropathic pain, a persistent change in the pain sensory system, is associated with reduced inhibition of nociceptive neurons. 5-HT and NE are implicated in modulating descending inhibitory pain pathways in the central nervous system, suggesting that the inhibition of both 5-HT and NE uptake may be essential for the reduction of persistent pain mechanisms [42]. Furthermore, the NE system may have more effective regulatory functions than the 5-HT system for allodynia [13,15,16]. Fibromyalgia and CIPN share some characters, such as pain hypersensitivity, sensorimotor deficits, debilitating effects, and uncleared pathophysiology [43]. The effect of milnacipran in fibromyalgia is estimated to increase the inhibitory neurotransmitter in descending pathways [12]. Therefore, milnacipran in cisplatin-induced peripheral neuropathy also could be anticipated to have similar effects in control of fibromyalgia.

Cisplatin and oxaliplatin are the chemotherapy drugs that include common in platinum agents. In our study, cisplatin injection was only intraperitoneally administrated. Andoh et al. reported the inhibitory action of milnacipran in oxaliplatin-induced mechanical allodynia according to different injection routes. Oxaliplatin-induced peripheral neuropathy was induced after single IP administration (3 mg/kg). Mechanical allodynia by oxaliplatin treatment presented highest allodynia score on day 10 after IP administration. Effect of oxaliplatin became insignificant by day 20. In this report, milnacipran significantly suppressed the oxaliplatin-induced mechanical allodynia when given systemically by IP injection dose-dependently (3–30 mg/kg). Intrathecal administration of milnacipran into the spinal cord was more effective with a lower dose than the systemic IP route (2.1–21 μg/site). But intracisternal and intracerebroventricular injections of milnacipran into the brain did not have an inhibitory effect on oxaliplatin-induced mechanical allodynia at the same doses. The inhibitory response of milnacipran in oxaliplatin-induced allodynia is likely regulated by acting on the spine rather than the brain [44]. Therefore, similarly, in cisplatin-induced mechanical allodynia, milnacipran is also presumed to have antihyperalgesic effects, mainly on the descending pathway of the spinal cord. And milnacipran, as a selective SNRI, might have an antihyperalgesic effect on cisplatin-induced neuropathy.

Cisplatin-induced neuropathy can be occurred through activation of nuclear factor kappa light chain enhancer of activated B cells (NF-kβ) pathway. The NF-kβ leads to release of pro-inflammatory cytokines and suppresses the release of anti-inflammatory cytokines by oxidative stress [45]. Meanwhile, the ATF3 is induced by stress and possesses a number of transcription factor binding sites including the NF-kβ [46]. Therefore, a Cisplatin-induced reduction in NF-kβ activation at the DRG could be considered to be another antihyperalgesic-presenting mechanism in cisplatin-induced neuropathy.

### 4.4. Limitations

There were some limitations in this research. First, the ATF3 levels in the DRG in this research were not measured according to different doses of milnacipran. The ATF3 levels in the DRG were evaluated only in the Milna50 group. The Milna10 group and Milna30 group were not evaluated in immunohistochemistry. Therefore, the change in ATF3 levels could not be identified according to the change in the milnacipran doses. And, the significant PWT change in the Milna50 group became meaningless after 24 h. But, we did not check the ATF3 levels in the DRG at each point of time slot in our study. The Milna10 and Milna30 groups might also be needed to measure ATF3 levels in DRG and compare the differences. If the PWT after administration of milnacipran decreases over time and mechanical allodynia symptoms reappear, it would be meaningful to measure the change by measuring the ATF3 level in the DRG after disappearing of milnacipran effect.

Second, Andoh et al. reported the antihyperalgesic effects of milnacipran in various administration routes such as systemic IP, intrathecal, intracerebroventricular, and intracisternal administration for oxaliplatin-induced mechanical allodynia [44]. But, in this study, the only milnacipran administration route was IP injection. In cisplatin-induced peripheral neuropathy, the mechanism and target sites of milnacipran could be identified and evaluated through various administration routes. A comparison by measuring doses of milnacipran that cause similar increases in PWT by administration routes will help to understand the principle of action of milnacipran in cisplatin-induced neuropathy.

Third, in this study, there was no drug to compare with milnacipran except for normal saline. In a prior study, we found gabapentin had a reversed effect in cisplatin-induced neuropathy. Antiallodynic pharmacology of gabapentin develops the effect by binding to the alpha2delta subunits of the calcium channel [22]. In addition to gabapentin, pregabalin and duloxetine, as medications for fibromyalgia, might be also needed to evaluate the antiallodynic effect in cisplatin-induced neuropathy.

Lastly, prevention of cisplatin-induced neuropathy was not evaluated in this study. In the prior study, pretreatment of etanercept was effective in delaying the onset of mechanical allodynia from cisplatin treatment [22]. In our study, the mice in experiment groups were already sufficiently neuropathic after IP cisplatin administration six times, on every other day, for a total dose of 13.8 mg /kg. During the early phase of allodynia in cisplatin chemotherapy, it will be necessary to investigate whether milnacipran can delay the development of mechanical allodynia.

## 5. Conclusions

Milnacipran administered via IP injection attenuated mechanical allodynia in mouse models of cisplatin-induced polyneuropathic pain. These effects were further corroborated by the significant suppression of neuronal ATF3 activation in the DRG. The efficacy and safety of milnacipran for cisplatin-treated patients should be clinically evaluated and further studied in the future.

## Figures and Tables

**Figure 1 pharmaceutics-15-02218-f001:**
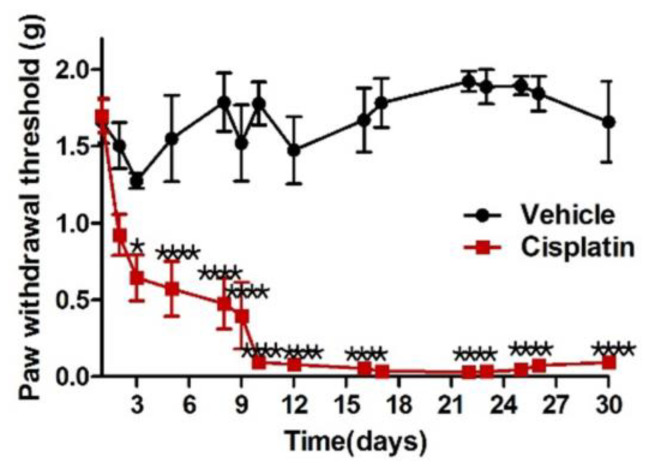
Cisplatin injection produces mechanical allodynia. * *p* < 0.05, **** *p* < 0.0001.

**Figure 2 pharmaceutics-15-02218-f002:**
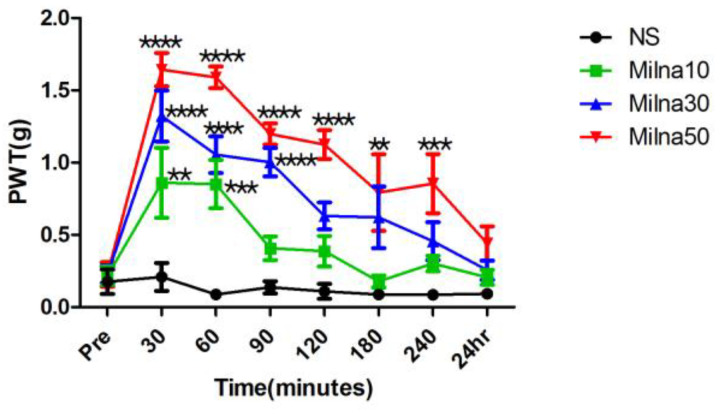
Cisplatin-induced mechanical allodynia is reversed by milnacipran. 24 h, 24 h; NS, normal saline; pre, prior to drug administration; PWT, paw withdrawal threshold. ** *p* < 0.01, *** *p* < 0.001, **** *p* < 0.0001.

**Figure 3 pharmaceutics-15-02218-f003:**
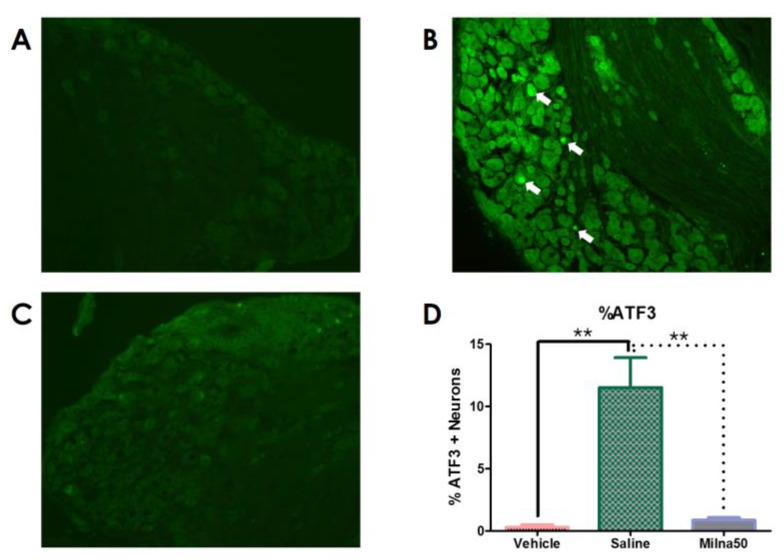
Milnacipran treatment in cisplatin-induced neuropathy decreased ATF3. Magnification; ×200. (**A**) Little ATF3 staining of the DRG of mice in the sham group (*n* = 6). (**B**) Cisplatin-treated mice exhibited higher activation of neuronal ATF3 (white arrow) (*n* = 6). (**C**) The ATF3 levels in the DRG were significantly lower in the Milna50 group (*n* = 6). (**D**) Milnacipran suppressed the activation of neuronal ATF3 in cisplatin-induced neuropathic mice. ATF3, activating transcription factor 3. ** *p* < 0.01.

## Data Availability

The data presented in this study are available on request from the corresponding author.

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
