# Peer review of "Milnacipran Has an Antihyperalgesic Effect on Cisplatin-Induced Neuropathy"

_pharmaceutics, 2023, doi:10.3390/pharmaceutics15092218_

Round 1
Reviewer 1 Report
Respect to neurons quantification. How many slides have been used to quantify the number of neurons? What are the microscope magnifications has the quantification been carried out? Has quantification been performed on the total DRG? It is not clear from the material and methods how this process has been carried out.
In relation to the treatment of animals with Minalcipran it is not clear when animals received the injection, I deduce that it is a single time, just after having given them cisplatin but when, the same day or how many days later? It must be clarified
A fundamental result of the article is the disappearance of immunoreaction for ATF in the DRG neurons, however the images presented in the work are of poor quality, moreover the magnifications not permit to check neuron positive inmunoreaction. I recommend that new images of better quality be added.
Author Response
Dear Reviewer: I wish to re-submit a review for publication in Pharmaceutics, titled “Milnacipran has an antihyperalgesic effect on cisplatin-induced neuropathy”. We thank you and the reviewers for your thoughtful suggestions and insights. The manuscript has benefited from these insightful suggestions. I look forward to working with you and the reviewers to move this manuscript closer to publication in Pharmaceutics. The manuscript has been rechecked and the necessary changes have been made in accordance with the reviewers’ suggestions. Our revisions in the manuscript are indicated in red. The responses to all comments have been prepared and attached herewith. Thank you for your consideration. I look forward to hearing from you. Sincerely, Hue Jung Park, MD, PhD. Department of Anesthesiology and Pain Medicine, Seoul St. Mary’s hospital, College of Medicine, The Catholic University of Korea, Seoul, Korea, 06591 Tel: +82-2-2258-6157, Fax: +82-2-537-1951 E-mail: huejung@catholic.ac.kr
Reviewer 2 Report
The manuscript seems superficial, although interesting. In the Introduction, the authors pay insufficient attention to the mechanism of cisplatin-induced allodynia. Accordingly, the choice of an serotonin-norepinephrine reuptake inhibitor for research in this particular model does not seem justified. Why exactly milnacipran with such a mechanism of action can work in this particular model?
Only one behavioral test was used to evaluate the effectiveness of the target compound, while no other substance was used to compare the effect. The fact that the authors point this out in the Discussion (section 4.4 Limitation) does not remove my question.
Only the levels of activating transcription factor 3 (ATF3) in the dorsal root ganglion was measured by immunohistochemistry. The authors need to justify why exactly this indicator was investigated in this model. the authors slightly touch on this issue in the Discussion, but, in my opinion, it is more appropriate to place information about the mechanism of development of chronic pain during cisplatin therapy and the role of ATF3 in the Introduction.
Moreover, the manuscript contains only data on the ATF3 level in cisplatin-treated mice and cisplatin-modeled mice treated with the substance. There is no information about the level of ATF3 in control mice without cisplatin administration. The photographs presented in the manuscript are of very poor quality and do not contain a scale ruler indicating the size in microns.
The authors point to the involvement of chronic inflammation in the development of cisplatin-induced neuropathy, but none of the widely used markers of inflammation has been investigated.
In general, in my opinion, this manuscript does not contain a sufficient amount of significant experimental data, and I would recommend changing the type of publication to a "short communication" or "brief report".
It is also necessary to correct the quotation in the text: "[]."
Author Response

(The authors gave the same response as above.)

Round 2
Reviewer 2 Report
The revised manuscript can be published as a Article.